# Puerto Ricans and Mexican immigrants differ in their psychological responses to patterns of lifetime adversity

Daniel K. Cooper[1]*, Mayra Bámaca-Colbert[2], Eric K. Layland[3], Emily G. Simpson[4], Benjamin L. Bayly[5]

1 Department of Psychology, University of South Carolina, Columbia, South Carolina, United States of America, 2 Department of Psychological Sciences (Developmental Area), University of California, Merced, California, United States of America, 3 Center for Interdisciplinary Research on AIDS, Yale School of Public Health, Yale University, New Haven, Connecticut, United States of America, 4 Department of Psychology, East Central University, Ada, Oklahoma, United States of America, 5 Agricultural Economics, Sociology, and Education, The Pennsylvania State University, University Park, Pennsylvania, United States of America

* dc47@mailbox.sc.edu

**Data Availability Statement:** We are unable to share the data used in this study because the data belongs to a third party. Our data use agreement explicitly prohibits the sharing of these data with

## Abstract

### Background

Puerto Ricans and Mexican immigrants are often exposed to multiple types of adversity across their lifetime (e.g., maltreatment, household dysfunction, discrimination) and this exposure can increase the risk for adult mental health problems.

### Purpose

The objective of this study was to (a) identify subgroups of individuals exposed to unique combinations of childhood adversity and lifetime discrimination among Puerto Ricans and Mexican immigrants, and (b) compare the prevalence of mental health problems across different risk profiles.

### Method

We used existing data from the HCHS/SOL Sociocultural Ancillary Study. Participants included Puerto Rican ($N$ = 402) and Mexican adults ($N$ = 1351) born outside but living in the continental U.S.

### Findings

Through latent profile analysis, we selected a three-profile solution for Puerto Ricans: (a) Low Exposure (low on all adversity items; 58% of sample), (b) Adverse Childhood Experiences (ACEs) Only (high on ACEs items, average or lower than average on discrimination items; 32%), and (c) Dual Exposure (high on all adversity items; 10%). For Mexicans, we selected a four-profile solution: (a) Low Exposure (52%), (b) ACEs Only (24%), (c) Maltreatment and Discrimination (15%), and (d) Dual Exposure (9%). For Mexicans, we found that the Dual Exposure and the Maltreatment and Discrimination profiles had the highest levels

non-approved users. However, data are available from BioLINCC (accession number: HLB01141418a) for researchers who meet the criteria for access to confidential data. Request for the data can be submitted using the following website: https://biolincc.nhlbi.nih.gov/studies/hchssol/.

**Funding:** This project was supported in the form of funding by the National Institute on Drug Abuse (P50 DA039838; PI: L. Collins) and the National Institute of Mental Health (T32 MH020031; PI: L. Kershaw) of the National Institutes of Health awarded to EL. The content is solely the responsibility of the authors and does not necessarily represent the official views of the National Institute on Drug Abuse or the National Institutes of Health. https://www.drugabuse.gov/. This project was also supported in the form of funding by the Prevention and Methodology Training Program (T32 DA017629; MPIs: J. Maggs & S. Lanza) with funding from the National Institute on Drug Abuse awarded to DC and ES. The content is solely the responsibility of the authors and does not necessarily represent the official views of the National Institute on Drug Abuse or the National Institutes of Health. https://www.drugabuse.gov/ The funders had no role in study design, data collection and analysis, decision to publish, or preparation of the manuscript.

**Competing interests:** The authors have declared that no competing interests exist.

of mental health problems. For Puerto Ricans, the Dual Exposure and ACEs Only profiles had the highest levels of mental health problems, suggesting that Puerto Ricans may be more vulnerable to the effects of childhood adversities as compared to Mexican immigrants. Results from our study indicate that different patterns of adversity exposure are linked to different levels of mental health outcomes, and therefore, may require different intervention dosage. Understanding which groups of individuals are at highest and lowest risk for mental health problems is critical for developing effective, tailored interventions to prevent the negative effects of childhood adversity and discrimination for Latinxs.

## Introduction

Lifetime adversity is an event or experience that causes psychological distress and increases the risk of developing physical and psychological problems, such as heart problems, cancer, smoking, depression, or posttraumatic stress disorder [1–3]. Past research has indicated that Latinx individuals are disproportionately affected by certain types of lifetime adversities as compared to non-Latinx whites [4]. A recent study, consisting of foreign-born and U.S.-born Latinxs living in the U.S., found that 77% of participants reported experiencing lifetime adversity during childhood (i.e., adverse childhood experiences; ACEs), and nearly 30% of these individuals experienced four or more adverse events [2]. These types of adversities can occur before, during, or after migrating to the continental U.S. Perceived discrimination, a powerful predictor of mental health, is another prevalent form of adversity Latinxs experience [5, 6]. However, few studies have examined the effects of unique combinations of childhood adversities and discrimination, despite their common co-occurrence. Even fewer studies have included Latinxs. To address this gap, the current study used a lifespan approach to identify and compare subgroups of Puerto Ricans and Mexicans born outside the continental U.S. at highest and lowest risk for mental health problems, based on their exposure to distinct patterns of adversity across the lifetime (i.e., childhood adversity and lifetime discrimination). This information is critical for improving the understanding of these co-occurring determinants of health and for developing interventions to interrupt their effects. For example, identifying subgroups at lowest and highest risk for various mental health problems can help inform decisions about who is at greatest need and who could benefit most from intervention. Analyzing Puerto Ricans and Mexicans separately allows for further nuance to our understanding of risk processes.

The *life course theory* [7] and *Latino critical theory* [8] guided our study's aims and hypotheses. Life course theory indicates that human development is shaped by and embedded in historical times and places. Similarly, we believe that a life course approach is needed to understand the effects of adversity across the lifespan [9]. Researchers assessing cumulative adversity with adults can benefit from examining adversity experienced in childhood and adulthood to fully capture the impact of lifetime adversity [10]. In accordance with life course theory, we examined the overlapping influence of two lifetime adversities (i.e., childhood adversity and lifetime discrimination) that have been shown to negatively impact the mental health and well-being of Latinxs [2]. Life course theory also guided our research hypotheses. For example, life course theory assumes that individuals' interpretations and responses to an event are influenced by their current context and past experiences. This aligns with our hypothesis that exposure to different patterns of risk factors would be linked with distinct mental health outcomes (discussed in greater detail below). Latino critical theory also informed our study hypotheses [8]. Latino critical theory is an extension of critical race theory

and examines the ways in which racism, sexism, classism, and other forms of oppression inter-act to impact people's lives. This theory (along with past research) informed our decision to include cultural stressors (e.g., discrimination) in our assessment of lifetime adversity. More-over, based on the numerous differences that exist between Puerto Ricans and Mexican immi-grants (e.g., immigration status, prevalence of discrimination), we assessed these groups separately to examine potential differences in exposure and response to lifetime adversity.

## Childhood adversity

Traditional measures of childhood adversity have assessed the incidence of numerous psycho-logically distressing events, such as child maltreatment (e.g., physical, sexual or emotional abuse, or neglect) and household dysfunction (e.g., divorce, living with a substance using par-ent, or a household member going to prison). The landmark ACEs Study [11] and subsequent research on childhood adversity resulted in several key findings. First, childhood adversity was relatively common [11, 12]. Second, adversities often co-occurred and had a graded relation-ship to numerous health outcomes, such as depression [12]. This means that the effects of childhood adversity became incrementally worse with each additional exposure to an adverse event (i.e., dose-response) [11, 13]. Third, subgroups of individuals often experienced specific patterns of adversity; and these exposure patterns influenced their risk for mental health prob-lems (e.g., dimensional and cluster-based approaches to adversity) [14–17]. For example, one study found that those who experienced child maltreatment were more likely to have mental health problems than those who experienced household dysfunction [18].

Several studies have used person-centered analysis (e.g., latent class analysis) to identify pat-terns of co-occurring ACEs that differ between subgroups. For example, Vaughn et al. [19] found that latent classes characterized by exposure to emotional and physical adversity ($n$ = 1262), family violence ($n$ = 358), and global adversity ($n$ = 246) were significantly more likely to have mood, anxiety, and substance use problems than the low adversity class ($n$ = 3,497). These studies provided evidence that lifetime adversities often cluster in distinct patterns and identifying these patterns may be useful for determining subgroups at risk for developing mental health problems.

Despite these advances in the field of childhood adversity, there are several limitations to this body of research. For one, prior literature has mainly focused on non-Latinx white sam-ples [11, 12, 20]. For example, 80% of participants in the landmark ACEs study were non-Latinx white [11]. This is an important gap to address because Latinxs are at a heightened risk compared to their non-Latinx white counterparts for several ACEs, such as maltreatment [4]. Research by Llabre et al. [2] represented one of the few efforts to assess the prevalence of ACEs among U.S. Latinx adults. They found that approximately 75% of Latinxs born outside of the continental U.S. reported experiencing at least one childhood adversity, which can be detri-mental to Latinxs' short and long-term health [21–23]. Second, the original ACEs scale did not include several types of adversity relevant to Latinxs (or other ethnically/racially minoritized groups), such as discrimination. Third, few studies have assessed lifetime adversity occurring outside of childhood. This is a key limitation because many children continue experiencing adversity in other life stages [10, 24]. For example, Mersky et al. [10] found that the effects of childhood adversity on adult mental health were fully mediated by adverse adult experiences. In other words, they found that the direct effect of childhood adversity on adult mental health was no longer significant after accounting for adult adversity. Moreover, prior research with military populations has found higher prevalence of ACEs among veterans as compared to civilians [e.g., 15], highlighting the co-occurring nature of adversities across the lifetime. Although this is a nascent area of research, these studies suggest that when examining early

risk factors for adult mental health, adult adversity should be assessed in addition to childhood adversity. This could be particularly important for Latinx populations, who often experience numerous stressors throughout their lifetimes (e.g., perceived discrimination, economic stress). Studying child adversity in isolation of adult adversity limits our ability to identify the collective influence of adversity across the lifespan.

## Perceived discrimination

Another form of lifetime adversity Latinxs and other ethnically/racially minoritized individuals commonly experience is perceived discrimination. Perceived discrimination refers to unfair treatment based on one's ethnic/racial background that is rooted in white supremacy and anti-black racism [25]. Latinxs can experience discrimination in a range of environments, such as at an interpersonal level (e.g., discriminatory comments) or a structural level (e.g., discriminatory policies). For the purposes of the present study, we focused on interpersonal discrimination that occurs over the course of one's life, which we refer to as perceived discrimination. For Latinxs born outside (but living in) the continental U.S., perceived discrimination can be related to anti-immigrant and/or anti-black beliefs. In 2019, nearly 60% of U.S. Latinxs reported experiencing perceived discrimination at least "from time to time," and even higher rates were reported for those with darker skin, highlighting the role of colorism and anti-blackness in these acts of discrimination [5]. Rates of perceived discrimination have increased in the last decade, in part, due to harsher immigration policies and increases in anti-immigrant sentiment in the U.S. [23, 26, 27]. For example, hate crimes have risen 80% in California since 2016 [28].

Studies have shown that experiencing perceived discrimination activates the body's stress response [29, 30] and contributes to negative mental health outcomes [31]. For example, Flores et al. [32] found that Mexican-origin adults' reports of perceived discrimination predicted depressive symptoms and poor general health, above and beyond the effects of general life stressors. Similarly, a study with Puerto Rican adults found that higher perceived discrimination was associated with higher depressive symptoms and perceived stress [6]. Although several recent studies have examined perceived discrimination among Latinx adults, prior studies assessing the effects of perceived discrimination on Latinxs have typically focused on youth [e.g., 33, 34] and have not considered co-occurring experiences of other forms of childhood adversity. Additional research is needed to advance the understanding of the effects of perceived discrimination across adulthood and in combination with other types of adversity.

## Latinx subgroup differences

Prior theory and research suggest that exposure and response to adversity can vary depending on context and culture [6, 35]. Latinxs are a heterogeneous group, made up of individuals with ethnic backgrounds from over 20 different countries. Therefore, it is reasonable to assume that the experiences of adversity may differ between Latinx subgroups. A study using the same dataset as in the current study, found differences in exposure to risk factors as well as differences in participants' responses to the risk factors between four Latinx ethnic groups born outside the continental U.S. (i.e., Mexicans, Puerto Ricans, Cubans, and Dominicans) [36]. Cooper et al. [36] reported significant differences in marital status, household income level, language preference, age, gender, and years lived in the continental U.S. They also found that certain risk factors were associated with mental health for some Latinx ethnic groups, but not for others. Specifically, discrimination was associated with mental health for Mexican immigrants, but not for Cuban, Puerto Rican, or Dominican immigrants. The present study aimed to build on Cooper et al.'s [36] study using a person-centered approach to determine whether

unique combinations of lifetime adversity were associated with distinct mental health outcomes for Latinxs born outside the continental U.S. Specifically, our goal was to conduct a detailed comparison of two Latinx groups to assess whether differences in exposure patterns to lifetime adversity influenced individuals' responses to adversities within these two groups.

Therefore, we selected Puerto Ricans and Mexicans born outside the continental U.S. because they represented some of variability that exists between U.S. Latinx groups (e.g., immigrant status, skin color, acculturation levels) and are the two largest Latinx ethnic groups living in the U.S. Although these groups share a common language, there are many within-group differences that may influence their perception and experience of adversity [37]. Few studies have conducted direct comparisons between Puerto Ricans and Mexicans born outside the continental U.S., nonetheless, past research comparing Latinx groups more broadly suggests that differences may exist. For example, several studies have suggested that Puerto Ricans have the highest rates of unemployment and one of the highest rates of divorce as compared to other Latinx groups [36, 38]. Puerto Ricans may also report one of the highest levels of discrimination [36, 39]. On the other hand, Mexican immigrants report higher rates of acculturation stress compared to other Latinx subgroups [40]. This could be attributed to having one of the lowest proportions of naturalized citizens, being more often targeted by immigration officials, and being least prepared to migrate to the U.S. as compared to other Latinx groups [38, 40].

In addition to Latinx ethnic group differences (i.e., between Latinx ethnic groups), there may be individual differences (i.e., within-group differences) that affect Latinxs' experience of adversity. Prior research suggests that experiences of discrimination may differ based on numerous individual factors which can vary between and within groups, such as immigration status [41], acculturation levels [41], ethnic identity [29, 42], and age [39]. Finch et al. [41], for instance, found that highly acculturated Mexican-origin adults reported higher levels of discrimination as compared to their less acculturated counterparts. Perez et al. [39] found that 25% of foreign-born Latinxs reported experiencing discrimination compared to 50% of U.S.-born Latinxs. Moreover, several studies have found that those with higher levels of ethnic identity reported experiencing less discrimination [39, 42]. In addition to isolating the effects of individual characteristics alone, it is important to understand the overlapping influences of these types of characteristics on health outcomes. One useful approach to identifying the ways in which different characteristics interact to influence response to risk is by using latent profile analysis. However, few studies have assessed subgroup differences based on Latinxs exposure to various patterns of risk. This information could be used to help determine subgroups at highest and lowest risk for mental health problems and inform preventive interventions tailored to the individuals' pattern of risk exposure.

## The present study

The aims of our study were to (a) identify latent profiles of Puerto Rican and Mexican individuals exposed to unique combinations of lifetime adversity (i.e., childhood adversity and perceived discrimination) and (b) compare the prevalence and level of adult mental health problems (i.e., trait anxiety, depressive symptoms, and trait anger) based on latent profile membership. We tested three main hypotheses. First, prior person-centered research on childhood adversity identified between 3–7 latent classes (or profiles) as being the best fit to the data [e.g., 14, 17, 19]. Therefore, we hypothesized that there would be at least three profiles of lifetime adversity. Second, based on our review of past literature and theory, we expected to find differences in the class structure and prevalence between Mexicans and Puerto Ricans. For example, we expected to find a lower prevalence of household dysfunction (e.g., divorce, neglect) for Mexicans [38]. Third, we hypothesized that we would find differences in the ways

in which certain risk profiles would be linked with mental health problems. For example, we expected that latent subgroups that experienced numerous ACEs in addition to discrimination would be at a higher risk for mental health problems than profiles with low adversity. Moreover, latent subgroups that experienced more harmful ACEs (e.g., abuse and neglect) [18] or more violent discrimination (e.g., threats) may be particularly vulnerable to mental health concerns.

## Method

### Sample

To address the aims of this study, we used the Hispanic Community Health Survey/Study of Latinos (HCHS/SOL) Sociocultural Ancillary Study (SCAS) dataset [43]. This ancillary study was conducted using a subset of participants recruited from the original HCHS/SOL parent study. The HCHS/SOL parent study randomly sampled households, using stratified probability sampling, in four U.S. cities with large Hispanic/Latinx populations, the Bronx, Miami, San Diego, and Chicago. Participants were eligible if they (a) self-identified as having Hispanic/Latinx background and (b) were between the ages of 18–74. The household level response rate was 33.5%. Of those who met the inclusion criteria, 41.7% consented to be in the study ($N$ = 16,415). The HCHS/SOL study was focused on risk and protective factors for health conditions and disease states (e.g., cardiovascular disease, diabetes) and the baseline assessments were conducted between 2008–2011.

In 2009, the SCAS was launched to assess psychosocial factors related to health conditions among U.S. Latinxs [43] using a representative sub-sample of the HCHS/SOL parent study ($N$ = 5313). The present study only included individuals who consented to sharing their data to investigators not associated with HCHS/SOL and associated laboratories ($N$ = 4645). We constrained our analytic sample to individuals born in the following two areas outside the continental U.S.: Puerto Rico ($n$ = 402) and Mexico ($n$ = 1351). These were two of the largest Latinx subgroups in the sample and represent the two largest Latinx subgroups living in the U.S. We obtained approval to conduct this secondary data analysis from the Institutional Review Board (STUDY00013534) at the Pennsylvania State University.

### Measures

**Lifetime adversity indicators.** Childhood adversity was assessed using the ACEs scale, a measure developed in a large-scale study by the Center for Disease Control and Prevention [11]. The ACEs scale examines exposure to various potentially stressful life events during the first 18 years of life. Participants reported whether they experienced the following 10 ACEs: physical abuse and neglect, emotional abuse and neglect, sexual abuse, witnessing abuse of caregiver, living with a substance user, living with a mentally ill person, parental divorce, and having a household member go to prison. Due to high collinearity, we combined the physical abuse and the emotional abuse items into one indicator so that 0 represented no physical/emotional abuse and 1 indicated that respondents answered "yes" to at least one of the two items. All other items were dichotomous with a score of one indicating that a participant reported experiencing a given adverse childhood experience. All items were included as binary indicators of childhood adversity in latent profile analysis (see Table 1).

We assessed perceived discrimination using the 17-item Brief Perceived Ethnic Discrimination Questionnaire-Community Version (PEDQ) [44]. The PEDQ assesses lifetime experiences of self-reported discrimination based on race or ethnicity. It includes various aspects of discrimination, including five stigmatization/evaluation items (e.g., "police officers have been unfair to you"), four threat/aggression items (e.g., "others threatened to hurt you"), four work/

**Table 1. Descriptive statistics of study variables for Mexicans and Puerto Ricans.**

| | Mexicans (*N* = 1351) | | | | Puerto Ricans (*N* = 402) | | | | *t*-test |
|---|---|---|---|---|---|---|---|---|---|
| | *M* | *SD* | α | Missing | *M* | *SD* | α | Missing | *p* |
| **Focal variables** | | | | | | | | | |
| Physical/emotional abuse | .43 | .49 | – | 2% | .36 | .48 | – | 2% | **.022** |
| Sexual abuse | .18 | .38 | – | 2% | .14 | .35 | – | 2% | **.042** |
| Emotional neglect | .24 | .43 | – | 2% | .22 | .42 | – | 2% | .366 |
| Physical neglect | .15 | .35 | – | 2% | .10 | .31 | – | 2% | **.017** |
| Parental divorce | .29 | .46 | – | 2% | .49 | .50 | – | 2% | **< .001** |
| Witnessed abuse | .22 | .41 | – | 2% | .29 | .46 | – | 2% | **.003** |
| Substance use | .33 | .47 | – | 2% | .38 | .49 | – | 2% | .054 |
| Mental illness | .15 | .36 | – | 2% | .25 | .43 | – | 2% | **< .001** |
| Prison | .22 | .41 | – | 2% | .28 | .45 | – | 2% | **.010** |
| Stigmatization/evaluation | 1.33 | .50 | .69 | 1% | 1.41 | .62 | .77 | 1% | **.022** |
| threat/aggression | 1.16 | .41 | .70 | <1% | 1.29 | .63 | .84 | 1% | **< .001** |
| Work/school | 1.55 | .67 | .69 | 1% | 1.60 | .78 | .75 | 1% | .186 |
| Exclusion/rejection | 1.85 | .78 | .74 | 1% | 1.91 | .86 | .78 | 1% | .233 |
| Depressive symptoms | 7.24 | 5.67 | .84 | 2% | 8.69 | 6.39 | .84 | 1% | **< .001** |
| Trait anxiety | 17.42 | 5.35 | .81 | 3% | 18.35 | 5.62 | .80 | 3% | **.004** |
| Trait anger | 16.41 | 4.93 | .84 | 2% | 16.98 | 5.81 | .86 | 1% | .075 |
| **Control variables** | | | | | | | | | |
| Age | 47.39 | 12.12 | – | 0% | 54.61 | 12.12 | – | 0% | **< .001** |
| Women | 65% | – | – | 0% | 59% | – | – | 0% | **.043** |
| Years in U.S. | 20.75 | 12.57 | – | 0% | 35.57 | 16.52 | – | <1% | **< .001** |
| Migration reason | | | | | | | | | |
| Economic reasons[a] | .50 | .50 | – | 1% | .30 | .46 | – | 1% | **< .001** |
| Came with parents | .14 | .35 | – | 1% | .31 | .46 | – | 1% | **< .001** |
| To be with family | .19 | .39 | – | 1% | .22 | .41 | – | 1% | .260 |
| Other | .17 | .38 | – | 1% | .17 | .38 | – | 1% | .969 |
| Prefer Spanish | 94% | – | – | 0% | 71% | – | – | 0% | **< .001** |
| Completed high school | 54% | – | – | <1% | 58% | – | – | 0% | .155 |
| Household Income <$30,000[b] | 65% | – | – | 5% | 76% | – | – | 7% | **< .001** |
| Health insurance coverage | 49% | – | – | <1% | 87% | – | – | 3% | **< .001** |
| Social support | 25.79 | 6.47 | .80 | 2% | 24.71 | 7.10 | .84 | 2% | **.004** |
| Ethnic identity | 3.52 | .46 | .65 | 6% | 3.72 | .46 | .67 | 3% | **< .001** |

*Notes.*

[a]Was treated as the reference group.

[b]Percent of individuals with family income less than $30,000. We presented migration reason, gender and income as percentages in the table to increase the ease of interpretation. Bold font indicates that variable means are significantly different between Mexicans and Puerto Ricans.

school items (e.g., "treated unfairly by coworkers or classmates"), and four exclusion/rejection items (e.g., "others made you feel like an outsider"). Respondents rated all items on a scale from 1 (*never*) to 5 (*very often*). To create scores for stigma/evaluation, exclusion/rejection, threat/aggression, and school/work discrimination, we calculated the mean item responses within each of the four dimensions of perceived discrimination so that higher scores reflected more discrimination. Mean scores were calculated for participants with complete data (i.e., responded to all 17 items). The PEDQ has been commonly used with Latinx populations [44, 45] and demonstrated acceptable internal consistency in this study (see Table 1). Each of the

four sum scores were included as continuous indicators of discrimination in the latent profile analysis.

**Mental health outcomes.**  Depressive symptoms were assessed using the 10-item Center of Epidemiologic Studies Scale (CES-D) [46]. The CES-D is a widely used measure that assesses respondents' levels of depressive symptoms experienced within the past week. Positive and negative items are rated on a scale from 0 (*rarely/less than one day*) to 3 (*all of the time/5-7 days*). Examples of positive items included: "I was happy" and "I felt hopeful about the future." Examples of negative items included: "I felt lonely" and "I could not get going." After reverse coding the positive items, we calculated scale scores by summing the 10 items (for those with complete data), with higher scores reflecting higher levels of depressive symptoms. Studies have provided support for the validity of the CES-D with Latinx populations [47]. See Table 1 for internal consistencies.

We assessed trait anxiety using a 10-item trait anxiety subscale from the State-Trait Anxiety Inventory [48]. The trait anxiety subscale (STAS) assessed aspects of anxiety that are thought to be relatively stable across time, such as level of general calmness. The STAS included six negative and four positive items relating to individuals' levels of general anxiety. Participants rated each item on a scale from 1 (*almost never*) to 4 (*almost always*). Examples of positive items included: "I feel satisfied with myself" and "I feel secure." Examples of negative items included: "I feel like a failure" and "I feel inadequate." For those with complete data, we calculated mean scores from the 10 items, with higher scores representing greater levels of trait anxiety. The STAS has been validated with various Latinx populations [49, 50]. See Table 1 for internal consistencies.

We assessed trait anger using the 10-item trait anger subscale from the State-Trait Anger Expression Inventory-2 (STAEI) [51]. The trait anger subscale assessed the aspects of anger that are thought to be relatively stable across time. Participants rated each item on a scale from 1 (*almost never*) to 4 (*almost always*). Example items included: "I am a hot-headed person," "I fly off the handle," and "when I get frustrated, I feel like hitting someone." For those with complete data, we summed the scores from the 10 items, with higher scores representing higher levels of trait anger. The STAEI has been validated with various Latinx populations [e.g., 52]. See Table 1 for internal consistencies.

**Covariates.**  We included several covariates found in past research to be associated with Latinx immigrants' mental health. We assessed age based on participants' self-reported age. We assessed gender based on participants' response to one item in which they indicated 0 (*woman*) or 1 (*man*). Years lived in the continental U.S. was assessed using one item in which participants indicated the length of time they had lived in the U.S. from the time of the interview. We assessed reason for migration using one item in which participants selected one of the following options as their main reason for migrating: 1 (*I moved with my parents as a child*), 2 (*to attend school*), 3 (*financial opportunity/work*), 4 (*refugee/political exile*), 5 (*to be with my family*), or 6 (*other*). We dummy coded this variable and collapsed options 2 (*to attend school*) and 4 (*refugee/political exile*) into the 6 (*other*) category due to having zero-inflated distributions. Language preference was determined using one item in which participants indicated the language they wanted to use for the interview: 0 (*Spanish*) or 1 (*English*). We assessed education level using one item in which participants' responses were grouped into one of three categories: 1 (*no high school diploma or GED*), 2 (*at most a high school diploma or GED*), 3 (*greater than high school or GED education*). Health insurance coverage was determined using one item in which participants indicated whether they possessed health insurance: 0 (*no current health insurance*), 1 (*currently have health insurance*). Social support was assessed using the sum of 12 items from the Interpersonal Support Evaluation List [53]. It assessed the availability of three types of social support: appraisal (advice or guidance), sense of belonging

(empathy or acceptance), and tangible (help or assistance), using four-point Likert-scale items, from 0 (*definitely false*) to 3 (*definitely true*). We assessed household income using one item in which participants were asked to report their household income: 1 (*less than $10,000*), 2 (*$10,001 - $20,000*), 3 (*$20,001 - $40,000*), 4 (*$40,001 - $75,000*), or 5 (*more than $75,000*). Finally, we assessed ethnic identity using a 12-item subscale from the Scale of Ethnic Experiences (SEE) [54]. Participants reported their thoughts and feelings regarding their ethnic group membership using a five-point Likert scale from 1 (*strongly disagree*) to 5 (*strongly agree*). For example, one item included, "I have a strong sense of myself as a member of my ethnic group." We created the scale score by calculating the mean of the 12 items, with higher scores reflecting higher ethnic identity.

**Data analysis plan.** Our data analysis included four steps: (a) preliminary analyses (e.g., descriptive statistics, assumption checks), (b) identifying latent profiles of lifetime adversity for Mexican immigrants and Puerto Ricans, (c) testing the association between latent profiles and demographic covariates and (d) testing the association between latent profiles and mental health outcomes (trait anger, trait anxiety, and depressive symptoms) controlling for age, gender, years lived in the continental U.S., reason for migration, language preference, household income, social support, and ethnic identity. All steps were conducted within a structural equation modeling (SEM) framework using Mplus 8 [55].

First, we conducted preliminary analyses to examine variable distributions, bivariate correlations and confirm the assumptions required for using structural equation modeling. Second, we used latent profile analysis (LPA) with mixed continuous and binary indicators to identify latent subgroups indicated by individual exposure to nine types of childhood adversity and four aspects of perceived discrimination. LPA is considered a person-centered approach for identifying subgroups (i.e., profiles) of individuals based on numerous observed characteristics [56]. This approach differs from standard variable-centered approaches that rely on arbitrarily grouping individuals based on an average or sum score on single measures. For example, prior ACEs literature has primarily classified ACEs exposure by the total number of adverse events individuals experienced or by subjectively placing individuals into "low," "medium," or "high" exposure groups. Conversely, LPA is a data-driven approach to identifying response patterns. Researchers can use LPA to determine the optimal number of groups that are the best fit to the data. This approach is particularly useful for assessing lifetime adversity because adverse events often co-occur in unique combinations. Different patterns of adversity may be linked to distinct health outcomes [57]. We determined the best fitting class structure by examining 1-profile to 7-profile models and examining the following model fit criteria: (a) smaller Bayesian information criterion (BIC) and sample size adjusted BIC values (a-BIC) [58], (b) smaller Akaike information criterion (AIC) values [59], (c) significant Lo-Mendell-Rubin (LMR) adjusted likelihood test [60], (d) a significant bootstrapped likelihood ratio test (BLRT) value [61], (e) higher entropy scores (approaching 1.00), and (f) profiles consisted of at least 5% of the sample, and (g) profiles were theoretically meaningful. Selecting the best fitting number of profiles and identifying profile structure was conducted independently for the subsets of Mexican and Puerto Rican participants.

Third, we assessed the association of demographic variables with profile membership using the modified three-step Mplus procedure (i.e., R3STEP auxiliary command) [62]. For latent profile analysis with covariates, this allows the estimation of the odds of belonging to each class compared to a reference class given the value or level of each covariate. Fourth, we assessed differences in mental health outcomes between the latent profiles using the three-step method introduced by Bolck, Croon, and Hagenaars (BCH) [63], controlling for age, gender, years lived in the continental U.S., reason for migration, language preference, household income, and ethnic identity. This approach accounts for uncertainty in modal class assignment (i.e.,

classification error) by incorporating classification error probabilities [64]. The BCH approach is recommended over one-step and classify- analyze approaches, because (a) subgroup definitions and sizes do not shift when the distribution of an outcome is misspecified and (b) estimates are less biased because participants are not assigned to a specific subgroup [65, 66]. Missing data ranged from 0–3% for the variables included in this analysis. We accounted for missing data using full-information maximum likelihood (FIML) estimation [67] in step b and listwise deletion in steps c and d.

## Results

### Preliminary analysis

We conducted preliminary analyses to assess variable distributions and examine bivariate correlations (see Table 1). For Mexicans, the average number of ACEs was 2.20. The most common ACEs for Mexicans were physical and emotional abuse (43%), caregiver substance use (33%), and parental divorce (30%). The highest scores for discrimination among Mexicans were on the exclusion/rejection ($M = 8.73$) and stigmatization/evaluation ($M = 6.66$) aspects. For Puerto Ricans, the average number of ACEs was 2.52. The most common ACEs were parental divorce (49%), caregiver substance use (38%) and physical and emotional abuse (37%). The highest discrimination scores were exclusion/rejection ($M = 9.22$) and stigmatization/evaluation ($M = 7.05$). Mexicans and Puerto Ricans had significantly different mean scores on most of the ACEs and perceived discrimination items. Correlations between the adversity items (ACEs and discrimination) ranged from .06 - .70 and most were statistically significant (see Tables 2 and 3).

### Latent profiles of lifetime adversity

We determined that a four-profile solution was the best fit to the data for Mexicans and a three-profile solution was the best fit for Puerto Ricans (see Table 4). Overall, the AIC, BIC, and a-BIC values improved as the number of profiles increased and the BLRT values were

**Table 2. Bivariate correlations of ACEs and perceived discrimination indicators for Puerto Ricans.**

|  | 1 | 2 | 3 | 4 | 5 | 6 | 7 | 8 | 9 | 10 | 11 | 12 | 13 |
|---|---|---|---|---|---|---|---|---|---|---|---|---|---|
| 1. Physical/ emotional abuse | – | | | | | | | | | | | | |
| 2. Sexual abuse | .319** | – | | | | | | | | | | | |
| 3. Emotional neglect | .463** | .318** | – | | | | | | | | | | |
| 4. Physical neglect | .294** | .151** | .257** | – | | | | | | | | | |
| 5. Parental divorce | .170** | .134** | .099* | .083 | – | | | | | | | | |
| 6. Witnessed abuse | .437** | .177** | .312** | .239** | .271** | – | | | | | | | |
| 7. Substance use | .387** | .123* | .182** | .247** | .137** | .371** | – | | | | | | |
| 8. Mental illness | .189** | .125* | .086 | .074 | .145** | .225** | .288** | – | | | | | |
| 9. Prison | .110* | .056 | .121* | .063 | .186** | .166** | .226** | .212** | – | | | | |
| 10. Stigma/ evaluation | .209** | .073 | .197** | .214** | .145** | .222** | .225** | .113* | .185** | – | | | |
| 11. Threat/ aggression | .205** | .072 | .192** | .094 | .040 | .211** | .181** | .074 | .091 | .507** | – | | |
| 12. Work/school | .251** | .065 | .266** | .223** | .164** | .237** | .203** | .105* | .114* | .537** | .456** | – | |
| 13. Exclusion/ rejection | .289** | .127* | .299** | .143** | .214** | .176** | .193** | .139** | .175** | .567** | .401** | .677** | – |

*Note.*

*$p < .05$.

**$p < .01$.

**Table 3.  Bivariate correlations of ACEs and perceived discrimination indicators for Mexicans.**

|  | 1 | 2 | 3 | 4 | 5 | 6 | 7 | 8 | 9 | 10 | 11 | 12 | 13 |
|---|---|---|---|---|---|---|---|---|---|---|---|---|---|
| 1. Physical/ emotional abuse | – | | | | | | | | | | | | |
| 2. Sexual abuse | .258** | – | | | | | | | | | | | |
| 3. Emotional neglect | .401** | .362** | – | | | | | | | | | | |
| 4. Physical neglect | .244** | .200** | .307** | – | | | | | | | | | |
| 5. Parental divorce | .171** | .115** | .167** | .165** | – | | | | | | | | |
| 6. Witnessed abuse | .394** | .184** | .265** | .271** | .161** | – | | | | | | | |
| 7. Substance use | .313** | .194** | .234** | .179** | .124** | .355** | – | | | | | | |
| 8. Mental illness | .181** | .167** | .214** | .089** | .095** | .125** | .243** | – | | | | | |
| 9. Prison | .182** | .121** | .117** | .087** | .100** | .140** | .224** | .241** | – | | | | |
| 10. Stigma/ evaluation | .231** | .101** | .166** | .132** | .089** | .084** | .124** | .134** | .154** | – | | | |
| 11. Threat/ aggression | .163** | .068* | .125** | .112** | .063* | .081** | .111** | .146** | .117** | .434** | – | | |
| 12. Work/school | .186** | .125** | .178** | .130** | .099** | .131** | .084** | .134** | .116** | .595** | .413** | – | |
| 13. Exclusion/ rejection | .211** | .113** | .209** | .143** | .072** | .131** | .117** | .152** | .142** | .586** | .394** | .640** | – |

*Note.*

*$p < .05$.

**$p < .01$.

significant for all profile solutions for both ethnic groups. For Mexicans, the three-, four- and five- profile solutions had comparable fit indices. We favored the four-profile solution over the three-profile solution because of the emergence of a distinct, conceptually meaningful fourth profile (i.e., ACEs Only) and over the five-profile solution because the fifth profile was small (less than 5% of the sample), which can lead to less reliable parameter estimates and limits the relevance of this subgroup [67, 68]. Moreover, the fifth profile was not conceptually distinct from the Dual Exposure profile. The six- and seven-profile models had profiles that were even less prevalent than the five-profile model and significant LMRT statistics, indicating worse fit. Therefore, for Mexicans, we selected the four-profile solution as the best fit to the data.

**Table 4.  Model fit statistics & selection criteria for Mexicans and Puerto Ricans.**

| Latinx Subgroup | #C | df | LL | AIC | BIC | a-BIC | Entr. | Smallest Class | LMR | BLR |
|---|---|---|---|---|---|---|---|---|---|---|
| **Mexicans (*n* = 1351)** | 1 | 489 | -11068.5 | 22171.1 | 22259.6 | 22205.6 | 1.00 | – | – | – |
| | 2 | 485 | -10037.2 | 20136.4 | 20297.8 | 20199.4 | .89 | 21.7% | < .001 | < .001 |
| | 3 | 474 | -9484.6 | 19059.2 | 19293.6 | 19150.6 | .85 | 8.9% | < .01 | < .001 |
| | **4** | **465** | **-9127.4** | **18372.9** | **18680.2** | **18492.8** | **.86** | **8.7%** | **< .001** | **< .001** |
| | 5 | 454 | -8768.1 | 17682.1 | 18062.3 | 17830.4 | .90 | 1.8% | > .05 | < .001 |
| | 6 | 445 | -8456.8 | 17087.6 | 17540.8 | 17264.4 | .89 | 1.8% | < .001 | < .001 |
| | 7 | 435 | -8234.6 | 16671.2 | 17197.3 | 16876.5 | .90 | 1.2% | > .05 | < .001 |
| **Puerto Ricans (*n* = 402)** | 1 | 499 | -3727.4 | 7488.8 | 7556.7 | 7502.8 | 1.00 | – | – | – |
| | 2 | 491 | -3364.6 | 6791.1 | 6915.0 | 6816.6 | .91 | 23.6% | < .01 | < .001 |
| | **3** | **480** | **-3190.4** | **6470.9** | **6650.7** | **6507.9** | **.87** | **10.3%** | **< .01** | **< .001** |
| | 4 | 470 | -3092.3 | 6302.6 | 6538.4 | 6351.2 | .89 | 5.7% | > .05 | < .001 |
| | 5 | 461 | -3015.9 | 6177.8 | 6469.5 | 6237.9 | .88 | 5.5% | > .05 | < .001 |
| | 6 | 451 | -2955.2 | 6084.3 | 6432.0 | 6156.0 | .89 | 2.2% | > .05 | < .001 |
| | 7 | 440 | -2908.3 | 6018.7 | 6422.3 | 6101.9 | .91 | 2.0% | > .05 | < .05 |

*Note.* #C = Number of Classes. df = Degrees of Freedom. LL = Log Likelihood. AIC = Akaike Information Criterion. BIC = Bayesian Information Criterion. a-BIC = adjusted BIC. Entr. = Entropy. LMR = Lo-Mendell-Rubin test. BLR = Bootstrapped Likelihood Ratio test. Bold font indicates the selected profile solution.

Table 5. Item response probabilities for Mexicans and Puerto Ricans.

| | Mexicans (N = 1351) | | | | | Puerto Ricans (N = 402) | | | |
|---|---|---|---|---|---|---|---|---|---|
| | Full Sample | 1 *Low Exp* (52%) | 2 *ACEs Only* (24%) | 3 M&D (15%) | 4 Dual Exp (9%) | Full Sample | 1 *Low Exp* (58%) | 2 *ACEs Only* (32%) | 3 Dual Exp (10%) |
| **ACEs Indicators** | | | | | | | | | |
| Phys/emo Abuse | .427 | .152↓ | .827↑ | .593↑ | .676↑ | .364 | .092↓ | .792↑ | .599↑ |
| Sexual abuse | .180 | .058↓ | .362↑ | .248 | .275↑ | .139 | .041↓ | .299↑ | .197 |
| Emotional neglect | .245 | .049↓ | .521↑ | .375↑ | .412↑ | .223 | .041↓ | .485↑ | .446↑ |
| Physical neglect | .147 | .045↓ | .275↑ | .231↑ | .242↑ | .103 | .019↓ | .229↑ | .197 |
| Divorce | .295 | .190↓ | .436↑ | .360 | .405↑ | .487 | .390↓ | .648↑ | .540 |
| Witnessed abuse | .215 | .028↓ | .544↑ | .269 | .319↑ | .291 | .109↓ | .522↑ | .618↑ |
| Substance use | .326 | .138↓ | .627↑ | .390 | .480↑ | .379 | .188↓ | .657↑ | .607↑ |
| Mental illness | .153 | .065↓ | .270↑ | .179 | .308↑ | .247 | .136↓ | .412↑ | .375 |
| Prison | .217 | .102↓ | .345↑ | .320↑ | .363↑ | .283 | .201↓ | .392↑ | .409 |
| **Perceived Discrimination Indicators** | | | | | | | | | |
| Stigmatization/ exclusion | 1.333 | 1.120↓ | 1.179↓ | 1.933↑ | 1.975↑ | 1.410 | 1.165↓ | 1.537 | 2.402↑ |
| Threat/ aggression | 1.161 | 1.032↓ | 1.058↓ | 1.100↓ | 2.325↑ | 1.287 | 1.077↓ | 1.159↓ | 2.859↑ |
| Work/school | 1.545 | 1.262↓ | 1.342↓ | 2.361↑ | 2.366↑ | 1.602 | 1.281↓ | 1.816↑ | 2.737↑ |
| Exclusion/ rejection | 1.851 | 1.503↓ | 1.691↓ | 2.746↑ | 2.780↑ | 1.908 | 1.528↓ | 2.251↑ | 2.994↑ |

*Note*. ACEs = Adverse Childhood Experiences. AND = Maltreatment and Discrimination. Low Exp = Low Exposure. Dual Exp = Dual Exposure. ↓Significantly below the group mean/probability. ↑Significantly above the group mean/probability.

For Puerto Ricans, we selected the three-profile solution as the best fit to the data. Although the AIC, BIC, and a-BIC values improved as the number of profiles increased, the size of the profiles were very small (less than or equal to 5%) and the improvement started to plateau after three profiles. Additionally, the four-profile solution did not have any conceptually distinct profiles as compared to the three-profile solution. Our decision was also based on significant LMRT statistics for the four-, five-, six- and seven-profile solutions, indicating that these solutions were a worse fit than the three-profile solution.

Three profiles were similar between Mexicans and Puerto Ricans (see Table 5). For Mexicans and Puerto Ricans, we labeled the profile that had lower than average scores on all the items, Low Exposure. We named the profile that had higher than average scores on the ACEs items, but average or lower than average on the discrimination items, ACEs Only. It is important to note that among Puerto Ricans, two aspects of discrimination (i.e., work/school and exclusion/rejection) were higher than average for the ACEs Only profile. However, we determined that the strength of the effect of one of these aspects of discrimination (work/school discrimination) was small 95% CI [-.19, -.02]. For this reason, we maintained that ACEs Only was an appropriate name for this profile. We named the profile that had higher than average scores on all childhood adversity and discrimination items, Dual Exposure. One profile was unique to Mexicans. We named this profile—which had higher than average scores on the abuse and neglect items, having a family member go to prison, and higher than average scores on three out of the four aspects of discrimination—Maltreatment and Discrimination. Our decisions regarding profile names were based on evaluating the indicator means and probabilities shown in Table 5.

We examined the characteristics of the risk profiles by testing covariates as predictors of profile membership using the R3STEP approach with the Low Exposure profiles set as the reference group (see Table 6). For Mexicans, we found that age, gender, reason for migration,

**Table 6. Results from LPA with covariates analysis for Mexicans and Puerto Ricans: Odd ratios.**

| | Mexicans (N = 1185) | | | | Puerto Ricans (N = 348) | | |
|---|---|---|---|---|---|---|---|
| | 1 Low Exposure (52%) | 2 ACEs Only (24%) | 3 M&D (15%) | 4 Dual Exposure (9%) | 1 Low Exposure (58%) | 2 ACEs Only (32%) | 3 Dual Exposure (10%) |
| Age | – | .98* | .97** | .96** | – | .98 | .96 |
| Woman | – | .60* | 1.23 | 2.10** | – | .87 | 2.49* |
| Years in U.S. | – | 1.01 | 1.01 | 1.02 | – | 1.00 | 1.02 |
| Migration reason | | | | | | | |
| Came with parents | – | .54* | .66 | .40* | – | 1.45 | 1.28 |
| To be with family | – | 1.00 | .74 | .74 | – | 1.00 | 1.07 |
| Other | – | 1.06 | 1.07 | 1.10 | – | 1.79 | .71 |
| Education level | – | 1.09 | .84 | .91 | – | .99 | .96 |
| Household income | – | .88 | .95 | .74* | – | .88 | 1.02 |
| Health insurance coverage | – | .85 | 1.00 | 1.25 | – | 1.13 | .43 |
| Prefer English | – | 1.61 | 3.35** | 2.63* | – | 1.72 | 1.31 |
| Social support | – | .98 | .93*** | .94** | – | 0.93** | .94* |
| Ethnic identity | – | .86 | 1.87** | 1.39 | – | .72 | .94 |

*Note.* Dashes indicate the reference profile. ACEs = Adverse Childhood Experiences. M&D = Maltreatment and Discrimination.

*$p < .05$.

**$p < .01$.

***$p < .001$.

household income, language preference, social support, and ethnic identity were significant predictors of profile membership relative to the Low Exposure profile. For example, as age increased, there were lower odds of belonging to the ACEs Only, Maltreatment and Discrimination, or Dual Exposure profile compared to the Low Exposure profile. As compared to women, men had greater odds of belonging to the Dual Exposure profile and lower odds of belonging to the ACEs Only profile compared to the Low Exposure profile. Mexicans who migrated with their parents as a child had lower odds of belonging to the ACEs Only or Dual Exposure profiles as compared to the Low Exposure profile. As household income level increased, the odds were lower for belonging to the Dual Exposure profile compared to the Low Exposure profile. As compared to those who preferred speaking Spanish, those who preferred speaking English had greater odds of belonging to the Dual Exposure and Maltreatment and Discrimination profiles as compared to the Low Exposure profile. As social support increased, the odds were lower for belonging to the Maltreatment and Discrimination and the Dual Exposure profile as compared to the Low Exposure profile. As ethnic identity increased, there were higher odds of belonging to the Maltreatment and Discrimination profile as compared to the Low Exposure profile.

For Puerto Ricans, gender and social support were significantly associated with risk profiles. Compared to women, men had greater odds of belonging to the Dual Exposure profile compared to the Low Exposure profile. As social support increased, the odds were lower for belonging to the ACEs Only and Dual Exposure profiles compared to the Low Exposure profile.

## Latent risk profiles and mental health

Using the BCH approach with regression, we found that mean levels of mental health problems (i.e., trait anxiety, depressive symptoms, trait anger) differed significantly across risk

**Table 7. Associations between latent profile membership and depressive symptoms, trait anxiety, and trait anger among Mexicans.**

|  | 1 β (95% CI) | 2 β (95% CI) | 3 β (95% CI) | 4 β (95% CI) |
|---|---|---|---|---|
| *Depressive symptoms* |  |  |  |  |
| Low Exposure | 14.15 (11.34, 16.96) | **1.96 (1.09, 2.84)** | **3.45 (2.49, 4.41)** | **3.83 (2.52, 5.14)** |
| ACEs Only | - | 16.11 (13.38, 18.85) | **1.49 (.34, 2.64)** | **1.87 (.43, 3.31)** |
| M&D | - | - | 17.60 (14.75, 20.44) | .38 (-1.11, 1.88) |
| Dual Exposure | - | - | - | 17.98 (15.06, 20.90) |
| *Trait anxiety* |  |  |  |  |
| Low Exposure | 26.70 (24.14, 29.26) | **1.57 (.76, 2.38)** | **3.29 (2.39, 4.20)** | **3.42 (2.27, 4.58)** |
| ACEs Only | - | 28.28 (25.78, 30.77) | **1.72 (.65, 2.79)** | **1.85 (.58, 3.12)** |
| M&D | - | - | 30.00 (27.40, 32.60) | .13 (-1.19, 1.46) |
| Dual Exposure | - | - | - | 30.13 (27.56, 32.69) |
| *Trait anger* |  |  |  |  |
| Low Exposure | 21.32 (18.86, 23.78) | **1.51 (.74, 2.28)** | **3.92 (2.96, 4.87)** | **4.01 (2.82, 5.20)** |
| Aces Only | - | 22.83 (20.31, 25.35) | **2.41 (1.25, 3.56)** | **2.50 (1.17, 3.82)** |
| M&D | - | - | 25.24 (22.72, 27.75) | .09 (-1.33, 1.51) |
| Dual Exposure | - | - | - | 25.32 (22.65, 28.00) |

*Note. n* = 1,171. Bold font indicates p < .05. Values on the diagonal indicate the mean value of each class as reference (i.e., the model intercept). Values above the diagonal indicate differences from the reference class associated with membership in each given class. All coefficients were adjusted for covariates (i.e., age, gender, years lived in the continental U.S., reason for migration, language preference, education level, household income, possession of health insurance, social support, and ethnic identity).

profiles. For Mexicans, the Dual Exposure and the Maltreatment and Discrimination profiles had significantly higher levels of all mental health problems than the ACEs Only and Low Exposure profiles. The Dual Exposure profile and the Maltreatment and Discrimination profile did not differ in their levels of mental health problems. For Puerto Ricans, the ACEs Only profile had significantly higher levels of all mental health problems as compared to the Low Exposure profile. The Dual Exposure profile had significantly higher levels of trait anger (but not trait anxiety or depressive symptoms) as compared to the Low Exposure profile. There was not a significant difference between the Dual Exposure and ACEs Only profiles in any of the mental health problems. See Table 7 and Table 8 for a full description of the differences in mental health outcomes between profiles.

## Discussion

This study used a lifespan approach [6] to assess two lifetime adversities commonly experienced by U.S. Latinxs, one prior to migration (i.e., ACEs) and one post migration (i.e., perceived discrimination). Ample research has linked childhood adversity [e.g., 20, 23] and perceived discrimination [e.g., 33, 37] with negative outcomes. However, few studies have examined the joint influence of childhood adversity and perceived discrimination with Latinx samples [3, 9]. Using LPA, we selected a four-profile solution for Mexicans and a three-profile solution for Puerto Ricans. Additionally, we found several differences in the ways in which the risk profiles were related to mental health problems for each Latinx ethnic group, after controlling for the effects of multiple variables shown to be associated with Latinx mental health (discussed below).

### Comparing risk profiles for Puerto Ricans and Mexicans

As expected, our best-fitting LPA solutions had at least three profiles. For Mexicans, we identified Low Exposure (52), ACEs Only (24%), Maltreatment and Discrimination (15%), and Dual

**Table 8. Associations between latent profile membership and depressive symptoms, trait anxiety, and trait anger among Puerto Ricans.**

| | 1 β (95% CI) | 2 β (95% CI) | 3 β (95% CI) |
|---|---|---|---|
| *Depressive symptoms* | | | |
| Low Exposure | 16.20 (9.80, 22.61) | **1.81 (.14, 3.48)** | 1.41 (-.74, 3.55) |
| ACEs Only | - | 18.01 (11.81, 24.22) | -.40 (-2.80, 1.99) |
| Dual Exposure | - | - | 17.61 (11.04, 24.19) |
| *Trait anxiety* | | | |
| Low Exposure | 29.40 (23.35, 35.45) | **2.16 (.60, 3.71)** | 1.69 (-.13, 3.51) |
| ACEs Only | - | 31.55 (25.71, 37.40) | -.47 (-1.69, 2.47) |
| Dual Exposure | - | - | 31.09 (24.88, 37.30) |
| *Trait anger* | | | |
| Low Exposure | 21.70 (14.99, 28.41) | **3.78 (2.20, 5.36)** | **3.16 (1.08, 5.23)** |
| ACEs Only | - | 25.48 (19.01, 31.95) | -.62 (-2.99, 1.75) |
| Dual Exposure | - | - | 24.86 (17.88, 31.84) |

*Note.* $n$ = 347. Bold font indicates $p < .05$. Values on the diagonal indicate the mean value of each class as reference (i.e., the model intercept). Values above the diagonal indicate differences from the reference class associated with membership in each given class. All coefficients were adjusted for covariates (i.e., age, gender, years lived in the continental U.S., reason for migration, language preference, education level, household income, possession of health insurance, social support, and ethnic identity).

Exposure profiles (9%). For Puerto Ricans, we identified Low Exposure (58%), ACEs Only (32%), and Dual Exposure profiles (10%). These results suggest that certain patterns of lifetime adversities tend to cluster together for Puerto Ricans and Mexican immigrants. Other studies using person-centered analysis to identify subgroups of individuals experiencing different combinations of adversity identified at least three profiles/classes [e.g., 14, 17, 19]. For example, one study with Latinx immigrants selected a four-class model for childhood adversity that included limited adverse experience (65%), emotional and physical abuse (24%), family violence (7%), and global adversity (5%) [19]. However, our study was the first to include perceived discrimination and ACEs items in the assessment of lifetime adversity.

We found three similarities in risk profiles between Mexicans and Puerto Ricans. First, both groups had ACEs Only profiles. This finding aligns with prior literature suggesting that ACEs are prevalent across U.S. Latinx populations [e.g., 2]. For example, Loria and Caughy [69] assessed the prevalence of ACEs in a national U.S. sample and found that 65% of Latinx immigrants reported at least one ACE and over 23% reported two or more ACEs. Second, both Latinx ethnic groups had a Low Exposure profile, in which individuals experienced lower than average levels of all lifetime adversity items. This aligns with past literature that has found low risk profiles when using person-centered analysis to examine patterns of adversity [e.g., 19]. Third, Puerto Ricans and Mexicans in our study each had a subgroup of individuals experiencing Dual Exposure (i.e., high levels of childhood adversity and high levels of perceived discrimination). Participants in our study may have experienced these adversities at different developmental periods (i.e., ACEs in childhood and perceived discrimination in adulthood) because most participants arrived in the U.S. after the age of 18. This means participants in our sample likely did not experience discrimination, at least not to the same extent, until arriving in the continental U.S. as adults. Several studies have found that childhood adversity is associated with experiencing adversity in adulthood [10, 70]. Our finding that subgroups of Puerto Ricans and Mexicans experienced dual exposure to ACEs and discrimination provides evidence for the importance of assessing cumulative adversity across different life

stages and is consistent with life course theory's emphasis on the developmental timing and historical context in assessing the impact of life events [35]. Future measures would benefit from adopting a lifespan approach to measuring adversity in order to better understand the experiences of U.S. Latinxs.

On the other hand, we found two main differences in profile structure and prevalence between Mexicans and Puerto Ricans. First, Mexicans had an additional profile (i.e., Maltreatment and Discrimination) that was unique to the Mexican sample. Puerto Ricans did not have a subgroup who were experiencing this combination of adversities. Second, we found that Puerto Ricans had higher mean discrimination scores in the Low Exposure, ACEs Only and Dual Exposure profiles as compared to Mexicans. This finding supported our hypothesis that there would be differences in the class structure and prevalence between Mexicans and Puerto Ricans. It also aligns with past literature that has found that Puerto Ricans may experience among the highest levels of discrimination as compared to other Latinx subgroups [39, 71].

## Risk profiles and mental health problems

As expected, results from BCH analyses indicated that risk profiles were significantly associated with Mexicans' and Puerto Ricans' mental health problems (i.e., depressive symptoms, trait anxiety, and trait anger), even after controlling for the effects of age, gender, years lived in the continental U.S., reason for migration, language preference, education level, household income, possession of health insurance, social support, and ethnic identity. Our findings correspond with past studies that found a link between lifetime adversities and mental health symptoms among Latinxs [21–23].

Additionally, results from BCH analyses indicated that many risk profiles varied in their levels of mental health problems (i.e., depressive symptoms, trait anxiety, and trait anger), after adjusting for control variables. For Puerto Ricans, individuals in the ACEs Only and Dual Exposure profiles had higher levels of mental health problems as compared to the Low Exposure profile. However, it was surprising that, for Puerto Ricans, the ACEs Only profile had similar levels of mental health problems as the Dual Exposure profile. This suggests that, for Puerto Ricans, experiencing discrimination in addition to childhood adversity is not more damaging than experiencing childhood adversity alone. This is contrary to the life course theory, which assumes that an accumulation of adversity across life stages is particularly damaging to health and well-being [32]. One potential explanation for this finding is that Puerto Ricans may be more resilient than Mexican immigrants to the effects of discrimination based on their citizenship status. In fact, Latino critical scholars have argued that individuals' experiences of the world are shaped by various intersecting aspects of their identity, such as language and citizenship status [8]. Studies have shown that anti-immigrant policy may drive discriminatory behaviors and attitudes towards Latinx immigrants [72]. However, Puerto Ricans may be more protected from these discriminatory processes than other Latinx groups because they are U.S. citizens by birth. Additionally, Puerto Ricans in our study had higher levels of health insurance coverage. Therefore, despite experiencing relatively high levels of discrimination, the impact on their mental health might be minimized given that being a U.S. citizen protects them from the fear of being deported or not hired due to their documentation status and allows for access to health insurance. These findings underscore the importance of adopting a critical, intersectional approach to understanding within- and between-group differences among Latinxs.

The relation between risk profiles and mental health problems was distinct for Mexicans. The Maltreatment and Discrimination and the Dual Exposure profiles had higher levels of mental health problems as compared to the ACEs Only and Low Exposure profiles. Said

differently, for Mexicans, the two risk profiles experiencing higher than average discrimination had higher levels of mental health problems as compared to the profile that experienced high levels of ACEs only. This suggests that Mexicans may be more vulnerable to the effects of discrimination as compared to ACEs. Accordingly, several scholars have argued that the literature examining the effects of childhood adversity on adult mental health often overlooks the mediational stressors that may occur along the way [10, 24]. For example, Roos et al. [24] found that adversity experienced in childhood is predictive of life adversity (i.e., homelessness) later in life. Moreover, Mersky et al. [10] found evidence that adult adversity was more predictive of adult mental health (i.e., anxiety, depression, and PTSD) than childhood adversity. These studies are consistent with life course theory which indicated that proximal risk factors may be more harmful than distal risk factors [32]. Similarly, our findings suggest that lifetime discrimination (likely experienced in adulthood) may be more harmful to Mexicans than childhood adversity alone.

## Implications

Our findings have implications for advancing the assessment of lifetime adversity and improving personalized prevention research aimed at tailoring mental health interventions to meet the specific needs of diverse populations. Individuals interested in assessing Latinx risk factors may benefit by including measures of perceived discrimination as well as childhood adversity, because certain subgroups may have been exposed to both types of adversity and could be at a heightened risk for negative outcomes. Understanding the effects of discrimination is particularly important in light of past research documenting the racialization of Latinx immigrants, a process that often associates Latinx immigrants with criminality [73]. Results from our study also suggest that different patterns of adversity exposure are linked to different levels of mental health outcomes, and therefore, may require different intervention dosage. For example, Mexicans experiencing abuse, neglect and discrimination or high ACEs and high discrimination may need a distinct and more intensive intervention than Mexicans exposed to high ACEs alone. Using a universal approach to prevention, in which everyone gets the same intervention, may be less effective in such cases where the needs of individuals may vary. Understanding which individuals are at the highest and lowest risk for mental health problems is critical for developing tailored programs to intervene across these risk levels. Consistent with Latino critical perspectives [8], our findings suggest that Latinx groups that experience the same pattern of risk factors may not have the same likelihood for developing mental health problems. This has important implications for intervention developers and policymakers focused on implementing prevention programs to promote resilience to adversity. We need to know which subgroups are at greatest risk for negative outcomes in order to effectively deliver interventions. Further studies are needed to substantiate our finding that experiencing unique patterns of lifetime adversity is associated with distinct mental health outcomes.

## Limitations

Our study was not without limitations. First, there were numerous demographic constraints to the study. For example, the sample we used from the HCHS/SOL SCAS data was comprised of Puerto Ricans and Mexicans born outside the continental U.S. living in well-established urban immigrant destinations (i.e., Chicago, Miami, San Diego, the Bronx), primarily middle-aged adults that have lived in the continental U.S. for an average of over 20 years. Therefore, our findings may not generalize to other Latinx populations outside of our sample (e.g., younger adults, recent migrants, individuals living in other areas of the U.S., continental U.S.-born individuals, Latinxs born in other parts of Latin America). Second, our measure of perceived discrimination did not specify the timing of the event. Therefore, we could not distinguish

those who experienced the event recently from those who experienced the event in the distant past. However, participants in our study indicated the frequency of the event, and it is likely that if they reported an event occurred "very often," than it was probably an ongoing experience. Third, our study assessed childhood adversity that occurred outside of the continental U.S., therefore, results are not generalizable to childhood adversity experienced in the U.S. Fourth, this was cross-sectional data, so we cannot infer temporality or causation. Fifth, we were unable to assess the effects of skin color variation or racial identity on the relation between discrimination and mental health outcomes, which past literature indicates to be an important factor to consider [8]. Sixth, although our measures had been validated with Latinx populations, few had been validated with our exact sample (i.e., Puerto Ricans and Mexicans born outside, but living in, the continental U.S.). However, this is a limitation to the broader field of mental health research, as more studies are needed to determine the validity of mental health measures across various Latinx populations. Finally, studies suggest that U.S. Latinx populations have experienced an increase in political stress due to the rise of anti-immigrant sentiment and policies [74]. Hence, experiences of discrimination may be different than they were at the time the data were collected.

## Conclusion

Our findings demonstrate the importance of using a lifespan approach to understand how lifetime adversity influences adult mental health among Latinx populations who experience additional adversities due to their minoritized status. Person-centered methods allowed us to examine how unique patterns of lifetime adversity were associated with Puerto Ricans' and Mexican immigrants' mental health. Initiatives to understand the long-term effects of childhood adversity could benefit from including measures of adversity that occurred in adulthood and that may be more prevalent experiences among certain populations to identify the developmental pathways and intermediate processes that may link childhood adversity to negative adult outcomes. Our findings provide additional evidence for the immense within-group variability of Latinx populations and highlight the need for personalized intervention approaches tailored to the unique needs of Latinx individuals. Understanding the unique patterns of risk factors that give rise to mental health problems is critical for developing effective interventions to promote resilience and protect against the negative effects of adversity.

## Author Contributions

**Conceptualization:** Daniel K. Cooper, Mayra Bámaca-Colbert, Emily G. Simpson, Benjamin L. Bayly.

**Formal analysis:** Daniel K. Cooper, Eric K. Layland.

**Methodology:** Daniel K. Cooper, Eric K. Layland.

**Resources:** Daniel K. Cooper.

**Writing – original draft:** Daniel K. Cooper.

**Writing – review & editing:** Daniel K. Cooper, Mayra Bámaca-Colbert, Eric K. Layland, Emily G. Simpson, Benjamin L. Bayly.

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
