## [Decision Letter · Decision Letter 0]

17 Nov 2020

PONE-D-20-27120

Identifying Subgroups of Latinxs Exposed to Lifetime Adversity: Links to Adult Mental Health Problems

PLOS ONE

Dear Dr. Cooper,

Thank you for submitting your manuscript to PLOS ONE. After careful consideration, we feel that it has merit but does not fully meet PLOS ONE’s publication criteria as it currently stands. Therefore, we invite you to submit a revised version of the manuscript that addresses the points raised during the review process.

We look forward to receiving your revised manuscript.

Kind regards,

Stephan Doering, M.D.

Academic Editor

PLOS ONE

Reviewers' comments:

Reviewer's Responses to Questions

**Comments to the Author**

1. Is the manuscript technically sound, and do the data support the conclusions?

Reviewer #1: Yes

Reviewer #2: Yes

2. Has the statistical analysis been performed appropriately and rigorously? 

Reviewer #1: Yes

Reviewer #2: Yes

3. Have the authors made all data underlying the findings in their manuscript fully available?

Reviewer #1: No

Reviewer #2: Yes

4. Is the manuscript presented in an intelligible fashion and written in standard English?

Reviewer #1: Yes

Reviewer #2: Yes

5. Review Comments to the Author

Reviewer #1: The authors investigated which subgroups with regard to childhood adversity and/ or ethnic discrimination exists in Mexican (N=1351) and Puerto Rican (N=402) immigrants living in the U.S.A.. Furthermore, they analyzed relations between the subgroups and indicators of mental health, i.e. depression, trait anxiety, and trait anger. The aim of the presented paper is of relevance for the field of mental health of immigrants, the selected methods are adequate, and the manuscript is well written.

I have the following (minor) suggestions to further improve the paper:

1. Introduction:

Please, be clearer about the difference between the present paper and the paper by the authors references as [33], since there may be some overlaps such as partially the same sample and depression as an outcome variable.

2. Methods:

Please, also present average scores for the PEDQ, since that would facilitate the appraisal of the scores.

It was unclear whether only immigrants were included or also persons that were born in the U.S. Please provide more information regarding this issue.

3. Tables:

I would suggest to add numbers to the first column in Table 2, in order to facilitate the attribution of the numbers found in the first line to the variables presented in the first column.

Reviewer #2: Please see attached comments that details my review and my comments for the authors.

This study examines dimensionality of adverse childhood experiences and perceived discrimination on the mental health risk among Mexican and Puerto Rican adults living in the US. The study offers a unique contribution in this area of research as the study, (1) assesses ACEs among rarely assessed Latinx-samples, specifically, (2) teases apart unique patterns of adversities versus past research that bundles all adversity together, (3) includes a rich dataset from the broader SOL study, and (4) conducts mental health research among Latinx populations that disaggregates versus typical studies that examine the group homogenously. The manuscript could benefit from much greater attention to the necessary theoretical background to this very important topic, as well as providing more clear, precise writing about the populations, patterns, and processes, etc. that are relevant to this study. Also, limitations of this study’s findings to the specific groups that are included in this study, and therefore the specific childhood contexts should be described, as these childhood contexts by and large exclude ACE-related experience occurring in the continental US for continental US-born/raised Latinx populations. I hope my comments below are useful to the authors in order to strengthen their manuscript further.

6. PLOS authors have the option to publish the peer review history of their article (what does this mean?). If published, this will include your full peer review and any attached files.

Reviewer #1: No

Reviewer #2: No

---

## [Author Response · Author response to Decision Letter 0]

13 May 2021

Editor’s Comments

1. Please ensure that your manuscript meets PLOS ONE's style requirements, including those for file naming. The PLOS ONE style templates can be found at https://journals.plos.org/plosone/s/file?id=wjVg/PLOSOne_formatting_sample_main_body.pdf and https://journals.plos.org/plosone/s/file?id=ba62/PLOSOne_formatting_sample_ title_authors_affiliations.pdf

RESPONSE: Thank you for these useful resources. We have revised the manuscript to meet the PLOS ONE style requirements. 

2. In your revised cover letter, please address the following prompts: (a) If there are ethical or legal restrictions on sharing a de-identified data set, please explain them in detail (e.g., data contain potentially sensitive information, data are owned by a third-party organization, etc.) and who has imposed them (e.g., an ethics committee). Please also provide contact information for a data access committee, ethics committee, or other institutional body to which data requests may be sent. (b) If there are no restrictions, please upload the minimal anonymized data set necessary to replicate your study findings as either Supporting Information files or to a stable, public repository and provide us with the relevant URLs, DOIs, or accession numbers. For a list of acceptable repositories, please see http://journals.plos.org/plosone/s/data-availability#loc-recommended-repositories. We will update your Data Availability statement on your behalf to reflect the information you provide.

RESPONSE: Thank you for these helpful directions. We have addressed these prompts in our cover letter. 

Reviewer #1’s Comments 

3. Introduction: Please, be clearer about the difference between the present paper and the paper by the authors references as [33], since there may be some overlaps such as partially the same sample and depression as an outcome variable.

RESPONSE: Thank you, great suggestion. We have added content to clarify the ways in which our study builds on our previous paper on p. 8, “The present study aimed to build on Authors’ [36] study using a person-centered approach to determine whether unique combinations of lifetime adversity were associated with distinct mental health outcomes for Latinxs born outside the continental U.S. Specifically, our goal was to conduct a detailed comparison of two Latinx groups to assess whether differences in exposure patterns to lifetime adversity influenced individuals’ responses to adversities within these two groups.”

4. Methods: Please, also present average scores for the PEDQ, since that would facilitate the appraisal of the scores. It was unclear whether only immigrants were included or also persons that were born in the U.S. Please provide more information regarding this issue.

RESPONSE: Based on further consideration of the literature and your comment, we chose to update the analysis using mean scores for each of the dimensions of perceived discrimination. 

We have revised the manuscript and added content to clarify our sample on p. 8, “The present study aimed to build on Authors’ [36] study using a person-centered approach to determine whether unique combinations of lifetime adversity were associated with distinct mental health outcomes for Latinxs born outside the continental U.S. Specifically, our goal was to conduct a detailed comparison of two Latinx groups to assess whether differences in exposure patterns to lifetime adversity influenced individuals’ responses to adversities within these two groups. Therefore, we selected Puerto Ricans and Mexicans born outside the continental U.S. because they represented some of variability that exists between U.S. Latinx groups (e.g., immigrant status, skin color, acculturation levels) and are the two largest Latinx ethnic groups living in the U.S.”

5. Tables: I would suggest to add numbers to the first column in Table 2, in order to facilitate the attribution of the numbers found in the first line to the variables presented in the first column. 

RESPONSE: We have added numbers to the first columns of Tables 2 and 3.

Reviewer #2’s Comments

1. The manuscript could benefit from much greater attention to the necessary theoretical background to this very important topic, as well as providing more clear, precise writing about the populations, patterns, and processes, etc. that are relevant to this study. 

RESPONSE: Thank you for this comment, we agree. We have revised the introduction and overall paper to improve the theoretical grounding for our study and the clarity and consistency of our terminology. We discuss the specific changes we made in our responses to comments below.

2. Also, limitations of this study’s findings to the specific groups that are included in this study, and therefore the specific childhood contexts should be described, as these childhood contexts by and large exclude ACE-related experience occurring in the continental US for continental US-born/raised Latinx populations. I hope my comments below are useful to the authors in order to strengthen their manuscript further.

RESPONSE: Thank you. We have clarified the populations to which our study is generalizable in the limitations section. Please see our responses to comments 23-25. Please let us know if you would like us to include any other information.

3. Abstract: As an example of adversity across the lifespan, “childhood adversity” reads as redundant and should be replaced with specific relevant adversities instead such as violence or poverty during childhood, interpersonal/structural discrimination across the lifespan, etc. 

RESPONSE: Thank you for this suggestion. We have revised the sentence as follows, “Puerto Ricans and Mexican immigrants are often exposed to multiple types of adversity across their lifetime (e.g., maltreatment, household dysfunction, discrimination) and this exposure can increase the risk for adult mental health problems.” 

4. It should be made clear whether mental health problems being assessed are lifetime, current, or adulthood only. 

RESPONSE: Thank you. We have revised the paper to have greater clarity around the timing of mental health problems. For example, we updated the first sentence of the abstract to focus on adult mental health problems. We also revised our aims statement on p. 9, “The aims of our study were to (a) identify latent profiles of Puerto Rican and Mexican individuals exposed to unique combinations of lifetime adversity (i.e., childhood adversity and perceived discrimination) and (b) compare the prevalence and level of adult mental health problems (i.e., trait anxiety, depressive symptoms, and trait anger) based on latent profile membership.”

5. For readers unfamiliar or with alternative views regarding labels, it is unclear the motivation for using Latinx in the Background when the study is restricted to mainland Puerto Rican and Mexican Latinx adults, specifically. Further, it is incorrect to refer to Puerto Rican and Mexican Latinx adults living in the continental US as ‘immigrants’ because (1) Puerto Rican adults living in the continental US are all US citizens at birth regardless whether they are born in the island or mainland US, and (2) Mexican adults living in the continental US may include a combination of US-born and immigrant populations. In general, the population of focus is unclear in the Abstract. 

RESPONSE: Thank you for this comment, we agree. We have updated the Abstract to reflect these suggestions (e.g., we revised the first sentence by replacing “Latinx” with “Puerto Ricans and Mexican immigrants”). We have also clarified our terminology regarding our sample (e.g., Puerto Ricans and Mexicans born outside (but living in) the continental U.S.) and we restrict our use of Latinx to discussions of the larger Latinx population.

6. Introduction: 1st paragraph, Page 3: It is unclear how prevalence of ACEs among Latinxs compare to other groups to assess whether 77% is more or less than average. It is also unclear what Latinx groups that this 77% includes – gender/sexual minorities, Mexican/Puerto Rican only as in the study, immigrant Latinx (non-US born), US-born only, etc. Do these adversities occur among US-born? 

RESPONSE: Thank you. We have revised the second sentence on p. 3 as follows, “Past research has indicated that Latinx individuals are disproportionately affected by certain types of lifetime adversities as compared to non-Latinx whites [4]. A recent study, consisting of foreign-born and U.S.-born Latinxs living in the U.S., found that 77% of participants reported experiencing lifetime adversity during childhood (i.e., adverse childhood experiences; ACEs), and nearly 30% of these individuals experienced four or more adverse events [2].”

7. Does perceived discrimination refer to interpersonal or structural as this construct has evolved considerably to tease apart different forms of discrimination and their impacts on health. Clearer definitions and statements should be provided according to the specific groups, terminology, and literature referenced. 

RESPONSE: This is an important suggestion. We have added content on p. 6 to address your comment, “Perceived discrimination refers to unfair treatment based on one’s ethnic/racial background that is rooted in beliefs about the racial superiority of certain groups (e.g., white individuals; 25]. Discrimination can occur in a range of environments, such as at an interpersonal level (e.g., discriminatory comments) or a structural level (e.g., discriminatory policies). For the purposes of the present study, we focused on interpersonal discrimination that occurs over the course of one’s life, which we refer to as perceived discrimination.”

8. 2nd paragraph, Page 3: It seems the LCT is insufficient to be the sole theory guiding this study. LCT guides the expected impacts of childhood exposures on outcomes in adulthood, including identification of sensitive periods, but doesn’t capture life course experiences/adversity that are specific to Latinx populations, that say critical race theory or weathering hypotheses could. Research by Maggie Alegria, William Vega, Hector Bird, etc. that have examined mental health epidemiology among Latinx populations specifically across the life course may support this background. 

RESPONSE: Thank you for this suggestion. We have incorporated Latino critical theory to supplement life course theory in addressing the importance of race/ethnicity on p. 4, “Latino critical theory also informed our study hypotheses [8]. Latino critical theory is an extension of critical race theory and examines the ways in which racism, sexism, classism, and other forms of oppression interact to impact people’s lives. This theory (along with past research) informed our decision to include cultural stressors (e.g., discrimination) in our assessment of lifetime adversity. Moreover, based on the numerous differences that exist between Puerto Ricans and Mexican immigrants (e.g., immigration status, prevalence of discrimination), we assessed these groups separately to examine potential differences in exposure and response to lifetime adversity.”

9. 3rd paragraph, Page 4-5: What is missing from childhood adversity is the validity of these measures to non-White groups thereby the focus of past research among non-white samples only. To what extent are Latinx sample included in ACE measurement studies? 

RESPONSE: Thank you for your comment, this is an important concern. We have added the following content on p. 5-6, “Second, the original ACEs scale did not include several types of adversity relevant to Latinxs (or other ethnically/racially minoritized groups), such as discrimination.”

10. Also, there is more recent theory about dimensionality of ACE’s effects on the brain, versus cumulative effects, that would be important to include in the background of this current study. See work by Katie A. McClaughlin et al., particularly PMID: 27773969. Finally, concerning adult adversity, literature about epigenetic effects could be included here as well as literature examining trauma which is much more developed and has demonstrated similar interactive effects between childhood ACE’s and traumas/adversities during adulthood and their effects on mental health (e.g., disaster-related and veterans-related research examining ACE’s plus adulthood adversities and their impacts on risk for post- traumatic stress). 

RESPONSE: Thank you for this important suggestion. We have incorporated the suggested reference on p. 4-5, “Third, subgroups of individuals often experienced specific patterns of adversity and these exposure patterns influenced their risk for mental health problems (e.g., dimensional and cluster-based approaches to adversity) [14-17].” Additionally, we added content regarding the co-occurring nature of adversity across the lifetime, such as in veteran samples: “Moreover, prior research with military populations has found higher prevalence of ACEs among veterans as compared to civilians [e.g., 15].”

11. 5th paragraph, page 5-6: The background on discrimination would benefit from describing the different kinds of discrimination (interpersonal/individual versus structural) and their impacts on health. Then it should be explicitly clear that this study examines individual only. It is also noted variation in experiences based on race/skin color within Latinx groups. To readers unfamiliar, it may be important to note the diversity within the Latinx group by race, ethnicity, language use, immigration status, geography, etc. that inform discrimination experiences. Also wouldn’t we expect intersectionality to matter here for Latinx who are black, immigrant, and/or gender/sexual minority such that interacting, multi-layered, co- occurring discrimination experiences present unique threats to health versus discrimination among white-passing, cis-gendered, US-born Latinx populations? Greater acknowledgement of these intersections can help move the knowledge base forward in this area (see original work by Kimberle Crenshaw for more information). 

RESPONSE: Thank you. We added content about discrimination and clarified our focus on interpersonal discrimination on p. 6, “Perceived discrimination refers to unfair treatment based on one’s ethnic/racial background that is rooted in beliefs about the racial superiority of certain groups (e.g., white individuals; 25]. Discrimination can occur in a range of environments, such as at an interpersonal level (e.g., discriminatory comments) or a structural level (e.g., discriminatory policies). For the purposes of the present study, we focused on interpersonal discrimination that occurs over the course of one’s life, which we refer to as perceived discrimination.” 

To be more explicit about diversity between and within Latinx groups, we added content in the first paragraph of p. 8, “The present study aimed to build on Authors’ [36] study using a person-centered approach to determine whether unique combinations of lifetime adversity were associated with distinct mental health outcomes for Latinxs born outside the continental U.S. Specifically, our goal was to conduct a detailed comparison of two Latinx groups to assess whether differences in exposure patterns to lifetime adversity influenced individuals’ responses to adversities within these two groups. Therefore, we selected Puerto Ricans and Mexicans born outside the continental U.S. because they represented some of variability that exists between U.S. Latinx groups (e.g., immigrant status, skin color, acculturation levels) and are the two largest Latinx ethnic groups living in the U.S. Although these groups share a common language, there are many within-group differences that may influence their perception and experience of adversity [37]. Few studies have conducted direct comparisons between Puerto Ricans and Mexicans born outside the continental U.S., nonetheless, past research comparing Latinx groups more broadly suggests that differences may exist.” We go on to discuss in more detail the within and between-group differences among Latinx groups.

To highlight the intersectional nature of individual characteristics with Latinx populations, we added two sentences in the last paragraph of p. 9, “In addition to isolating the effects of individual characteristics alone, it is important to understand the overlapping influences of these types of characteristics on health outcomes. One useful approach to identifying the ways in which different characteristics interact to influence response to risk is by using latent profile analysis.”

12. Concerning Latinx heterogeneity, this text should be introduced much earlier to help guide the reader as to why we would expect variation in the application of ACEs and LCT in research pertaining to Latinx populations. 

RESPONSE: Thank you for this comment. We have added the following content on p. 4, “Moreover, based on the numerous differences that exist between Puerto Ricans and Mexican immigrants (e.g., immigration status, prevalence of discrimination), we assessed these groups separately to examine potential differences in exposure and response to lifetime adversity.”

13. Also, it is unclear why Mexican and Puerto Rican groups are of focus – is only because they are the largest sub-groups or is it that other groups are underrepresented in research? How does research on Mexican and Puerto Rican populations generalize to other groups? What are the public mental health impacts of focusing on the two largest sub-groups only? How do these two groups differ in terms of other socio-demographics such as race, age, immigration status, language use, etc. (PEW Hispanic Research Center may provide this information). Why determine highest risk groups for tailored intervention versus low-dose, ubiquitous interventions for all (e.g., universal mental health coverage, Spanish-language competent providers, etc.) – what are the tradeoffs of one approach versus another? These motivations should be made clearer. 

RESPONSE: These are important comments, thank you. We have added content to clarify the justification for our sample. See our response to comment #11.

We agree that it is important to identify those at highest and lowest risk to inform targeted intervention approaches. We clarified this on p. 9, “This information could be used to help determine subgroups at highest and lowest risk for mental health problems and inform preventive interventions tailored to the individuals’ pattern of risk exposure,” as well as in several other places in the manuscript.

We added the following content to discuss the universal vs. tailored approaches on p. 30, “Results from our study also suggest that different patterns of adversity exposure are linked to different levels of mental health outcomes, and therefore, may require different intervention dosage. For example, Mexicans experiencing abuse, neglect and discrimination or high ACEs and high discrimination may need a distinct and more intensive intervention than Mexicans exposed to high ACEs alone. Using a universal approach to prevention, in which everyone gets the same intervention, may be less effective in such cases where the needs of individuals may vary. Understanding which individuals are at the highest and lowest risk for mental health problems is critical for developing tailored programs to intervene across these risk levels.”

14. Methods: Sample, Page 9: It is unclear the motivation for restricting the sample to two Latinx groups only. It is also unclear from the Abstract and Introduction if this study and Background is pertaining to Puerto Rican and Mexican adults living in the mainland US who were not born in the mainland US, specifically. The only reason provided, “These were two of the largest Latinx subgroups in the sample and represent the two largest Latinx subgroups living in the U.S.” is still unclear in terms of what is meant by “subgroup”. Ethnic subgroup? Among immigrant populations? Among non-mainland US born populations? These definitions and rationale for inclusion criterion are unclear yet they should be explicit throughout the manuscript. 

RESPONSE: Thank you. See our response to the previous comment #11. Additionally, we revised to manuscript to use greater precision and consistency in our terminology describing our sample (e.g., “Puerto Rican and Mexicans born outside the continental U.S.”). 

15. Also, if 1753 of the 4645 were included only, how did the excluded sample compare to those that were included? Why were they excluded and why not assess South American and Central American and Caribbean groups separately? 

RESPONSE: Please refer to our response to comment #11 for a justification for our sample. We also felt that treating Latinxs from South America or Central America as two large groups overlooks the immense variability that exist within these populations. 

16. Page 9-10: What is the validity of the ACE, discrimination, and mental health measures among the included Latinx populations in this study, specifically? That is, Puerto Rican and Mexican adults living in the mainland US who were not born in the mainland US. Also, alpha reliability of each measure would be helpful to include in the text. 

RESPONSE: Thank you for this comment. These measures have been used with U.S. Latinx populations broadly (i.e., majority Mexican samples), but few have assessed the validity within exclusively Mexican or exclusively Puerto Rican samples. We have included the following text in the limitations section on p. 32, “Sixth, although our measures had been validated with Latinx populations, few had been validated with our exact sample (i.e., Puerto Ricans and Mexicans born outside, but living in, the continental U.S.). However, this is a limitation to the broader field of mental health research, as more studies are needed to determine the validity of mental health measures across various Latinx populations.” 

It seems somewhat repetitive to have the reliabilities of each scale for Puerto Ricans and Mexicans listed in the text as well as in Table form. In our opinion, the Table presents a clear overview of the internal consistencies as well as other descriptive statistics. 

17. Are there covariates available to assess racial/skin color variation as this would be highly relevant to the discrimination measures? Also, potential buffers or resilience factors such as health insurance status, education, and close relationships that mitigate risks of adversity/discrimination on mental health should be included. Further, is state/geography examined as CA/NY present more immigrant-favorable policies versus FL, etc. (e.g., Hatzenbuehler et al. study cited regarding state immigration policies and mental health). 

RESPONSE: The one race item in the dataset was not a useful covariate for the present sample, as 50% of Mexican immigrants did not respond (i.e., either refused or selected the unknown/no response option) and about 40% of Puerto Ricans did not respond. This indicates that the survey items did not adequately capture many participants’ conceptualizations of their racial identity, and therefore, we were unable to reliably assess the impact of racial identity or skin tone. 

Based on your suggestions, we added health insurance status, education level, and social support as covariates and updated our analyses. 

We have revised our limitations section to clarify the samples to which our study findings can be generalized. For example, on p. 30, we stated, “First, there were numerous demographic constraints to the study. For example, the sample we used from the HCHS/SOL SCAS data was comprised of Puerto Ricans and Mexicans born outside the continental U.S. living in well-established urban immigrant destinations (i.e., Chicago, Miami, San Diego, the Bronx), primarily middle-aged adults that have lived in the continental U.S. for an average of over 20 years. Therefore, our findings may not generalize to other Latinx populations outside of our sample (e.g., younger adults, recent migrants, individuals living in other areas of the U.S., continental U.S.-born individuals, Latinxs born in other parts of Latin America).”

18. Discussion: Page 25-26: US citizenship status comes with greater access to health insurance as well, therefore, access to mental health care, which can mitigate negative effects of discrimination on mental health among Puerto Ricans. In contrast, those without access or without a public insurance option may be more prevalent among Mexican immigrants, particularly those that are undocumented. Nevertheless, racial/skin color variation isn’t taken into account here when describing differences between Puerto Rican and Mexican discrimination experiences. 

RESPONSE: These are great comments, thank you. We have revised the end of the first paragraph on p. 29 as follows, “Additionally, Puerto Ricans in our study had higher levels of health insurance coverage. Therefore, despite experiencing relatively high levels of discrimination, the impact on their mental health might be minimized given that being a U.S. citizen protects them from the fear of being deported or not hired due to their documentation status and allows for access to health insurance.” 

In regards to skin color variation, we have included content in our limitations section: “Fifth, we were unable to assess the effects of skin color variation or racial identity on the relation between discrimination and mental health outcomes, which past literature indicates to be an important factor to consider [8].”

19. Limitations of the sample should be elaborated as this is a highly specific sample. There are many unique Latinx groups excluded from this study including, but not limited to, US-born Latinx, Gen Z and Millennial Latinx, gender/sexual minorities (gender assessed as binary variable), no racial variation captured, rural/suburban Latinx populations (different from new settlement areas as indicated), indigenous Latinx, as well as Central/South American and Other Caribbean groups. That US-born, young Latinx are excluded limits generalizability of these findings to understanding ACE’s that occur in the continental US context, specifically. The ACE’s captured in this study reflect those that occur in Mexico or the island of Puerto Rico, and not childhood adversities that are experienced in the continental US for other Latinx populations. Further, time spent back and forth between Mexico or Puerto Rico and mainland US during childhood, especially for Puerto Ricans who are citizens and arguably have easier migration abilities, but also Mexican border populations, isn’t factored into this study. These experiences could introduce a unique set of migration-related ACE’s that are not captured by the ACE measure. These important limitations should be noted. These exclusions to the sample should also be noted, and as such, use of Latinx should be re-considered or defined.

RESPONSE: Thank you, we agree. We have revised the limitation section to reflect these considerations. See our response to comments 17-18.

As described above, we have clarified our terminology regarding our sample (e.g., Puerto Ricans and Mexicans born outside (but living in) the continental U.S.) and we restrict our use of Latinx to discussions

---

## [Decision Letter · Decision Letter 1]

2 Aug 2021

PONE-D-20-27120R1

Puerto Ricans and Mexican immigrants differ in their psychological responses to patterns of lifetime adversity

PLOS ONE

Dear Dr. Cooper,

Thank you for submitting your manuscript to PLOS ONE. After careful consideration, we feel that it has merit but does not fully meet PLOS ONE’s publication criteria as it currently stands. Therefore, we invite you to submit a revised version of the manuscript that addresses the points raised during the review process.

We look forward to receiving your revised manuscript.

Kind regards,

Stephan Doering, M.D.

Academic Editor

PLOS ONE

Journal Requirements:

Reviewers' comments:

Reviewer's Responses to Questions

**Comments to the Author**

1. If the authors have adequately addressed your comments raised in a previous round of review and you feel that this manuscript is now acceptable for publication, you may indicate that here to bypass the “Comments to the Author” section, enter your conflict of interest statement in the “Confidential to Editor” section, and submit your "Accept" recommendation.

Reviewer #1: (No Response)

Reviewer #3: (No Response)

2. Is the manuscript technically sound, and do the data support the conclusions?

Reviewer #1: Yes

Reviewer #3: Partly

3. Has the statistical analysis been performed appropriately and rigorously? 

Reviewer #1: Yes

Reviewer #3: N/A

4. Have the authors made all data underlying the findings in their manuscript fully available?

Reviewer #1: (No Response)

Reviewer #3: Yes

5. Is the manuscript presented in an intelligible fashion and written in standard English?

Reviewer #1: Yes

Reviewer #3: Yes

6. Review Comments to the Author

Reviewer #1: The authors did a great work in thoroughly revising the manuscript. All my comments were addressed. However, I have two further (minor) suggestions with regard to the revised analyses:

1. Table 6: While the text says that "As compared to those who preferred speaking Spanish, those who preferred speaking English had greater odds of belonging to the Dual Exposure and Maltreatment and Discrimination profiles as compared to the Low Exposure profile." the respective line in Table 6 is named "Prefer Spanish". That is confusing and I would suggest to change the name of the line in "Prefer English". Moreover, I wondered whether the categorization of Spanish=0 and English =1 is appropriate for the analyses of Odds ratios, since -to the extent I know- no meaningful Odds ratio can be determined for a category 0.Please consider changing the categorizations using the values 1 and 2.

2. Results on the associations between latent risk profiles and mental health in Puerto Ricans: The non-significance of the (possible) differences for the Dual Exposure group may be a result of the small sample size of this sub-group (with n=35?) and the lack of statistical power. Please discuss this possibility.

Reviewer #3: Thank you for the great work presented in this paper. I had a few suggestions for the manuscript to help improve some of the essential contributions this paper offers. The introduction needs to have a more straightforward argument because there is not enough emphasis on this paper's contribution from the start. Additionally, the childhood adversity section should explicitly state why Latinx people are impacted differently in adulthood and why this outcome matters. The paper remains vague in this section of the literature. The perceived discrimination section lacks the critical race framework that exists in the literature regarding the racialization of Mexican and Puerto Ricans. These perceived discriminatory practices also referred to as racial microaggressions, are rooted in the anti-blackness in the United States. It is also essential to discuss the colorism that plays a role in these micro-aggressions. For some Latinx people with lighter complexions, discrimination does not occur until they leave their home country. Further, the result and implications need to discuss the racialization of Latinx people because these sections need to connect the critical race frameworks that exist in Latinx studies. Overall, the results and methodological approach of this study are intriguing and important.

7. PLOS authors have the option to publish the peer review history of their article (what does this mean?). If published, this will include your full peer review and any attached files.

Reviewer #1: No

Reviewer #3: No

---

## [Author Response · Author response to Decision Letter 1]

10 Sep 2021

Reviewers' comments

Reviewer #1

1. Table 6: While the text says that "As compared to those who preferred speaking Spanish, those who preferred speaking English had greater odds of belonging to the Dual Exposure and Maltreatment and Discrimination profiles as compared to the Low Exposure profile." the respective line in Table 6 is named "Prefer Spanish". That is confusing and I would suggest to change the name of the line in "Prefer English". Moreover, I wondered whether the categorization of Spanish=0 and English =1 is appropriate for the analyses of Odds ratios, since -to the extent I know- no meaningful Odds ratio can be determined for a category 0. Please consider changing the categorizations using the values 1 and 2.

RESPONSE: We appreciate this comment. We have revised the variable in Table 6 to “Prefer English,” thank you for catching this. Regarding the second aspect of this comment, our understanding is that the values assigned to each group (e.g., “0” and “1” vs. “1” and “2”) does not matter, they are merely to distinguish the two groups from one another for the purposes of analysis and would not change the significance or effect sizes of the results. We are happy to provide more background on this if the reviewer requests it.

2. Results on the associations between latent risk profiles and mental health in Puerto Ricans: The non-significance of the (possible) differences for the Dual Exposure group may be a result of the small sample size of this sub-group (with n=35?) and the lack of statistical power. Please discuss this possibility.

RESPONSE: Thank you for this comment. We agree that the Dual Exposure profile’s smaller sample size limited power to compare across profiles. However, the findings we highlighted in the discussion section focused on the fact that the ACEs Only Profile had similar (and actually higher levels of mental health problems, although not reaching significance) levels of mental health problems as compared to the Dual Exposure Profile. This was contrary to our hypothesis that experiencing a combination of discrimination and childhood adversity would be associated with the greatest mental health problems. We believe that adding content describing the non-significant differences we found between the Dual Exposure and Low Exposure profiles (for Puerto Ricans) in the discussion section would be less interesting and relevant for readers. 

Reviewer #3

3. The introduction needs to have a more straightforward argument because there is not enough emphasis on this paper's contribution from the start. 

RESPONSE: Thank you for this comment. We have added content on the first page to make the goal and significance of this paper clearer from the outset, “This information is critical for improving the understanding of these co-occurring determinants of health and for developing interventions to interrupt their effects. For example, identifying subgroups at lowest and highest risk for various mental health problems can help inform decisions about who is at greatest need and who could benefit most from intervention. Analyzing Puerto Ricans and Mexicans separately allows for further nuance to our understanding of risk processes.”

4. Additionally, the childhood adversity section should explicitly state why Latinx people are impacted differently in adulthood and why this outcome matters. The paper remains vague in this section of the literature. 

RESPONSE: We appreciate this comment. To clarify, our argument is not that the long-term effects of childhood adversity are different for Latinx vs. non-Latinx groups, rather, that exposure to childhood adversity in combination with adult adversity is important to assess in this population because of the high prevalence of various lifetime adversities. We added content to clarify this on p. 6, “This could be particularly important for Latinx populations, who often experience numerous stressors throughout their lifetimes (e.g., perceived discrimination, economic stress). Studying child adversity in isolation of adult adversity limits our ability to identify the collective influence of adversity across the lifespan.”

5. The perceived discrimination section lacks the critical race framework that exists in the literature regarding the racialization of Mexican and Puerto Ricans. These perceived discriminatory practices also referred to as racial microaggressions, are rooted in the anti-blackness in the United States. It is also essential to discuss the colorism that plays a role in these micro-aggressions. For some Latinx people with lighter complexions, discrimination does not occur until they leave their home country. 

RESPONSE: Thank you. We added content in this section to highlight the role of anti-black racism and colorism in Latinxs’ experiences of perceived discrimination: “For Latinxs born outside (but living in) the continental U.S., perceived discrimination can be related to anti-immigrant and/or anti-black beliefs. In 2019, nearly 60% of U.S. Latinxs reported experiencing perceived discrimination at least “from time to time,” and even higher rates were reported for those with darker skin, highlighting the role of colorism and anti-blackness in these acts of discrimination [5].”

6. Further, the result and implications need to discuss the racialization of Latinx people because these sections need to connect the critical race frameworks that exist in Latinx studies. 

RESPONSE: We appreciate the reviewer’s continued attention to contextualizing the study findings within critical perspectives, and we agree this could be more explicit. In addition to the content we added in the previous revision (e.g., “In fact, Latino critical scholars argued that individuals’ experiences of the world are shaped by various intersecting aspects of their identity, such as citizenship status [8].”), we added additional content in the discussion and implications section to address this feedback. For example, in the second to last paragraph in the discussion section we added, “These findings underscore the importance of adopting a critical, intersectional approach to understanding within- and between-group differences among Latinxs.” In the implications section, we added, “Understanding the effects of discrimination is particularly important in light of past research documenting the racialization of Latinx immigrants, a process that often associates Latinx immigrants with criminality [72].” Please let us know if you would like to see additional content included.

---

## [Editor Report · Decision Letter 2]

27 Sep 2021

Puerto Ricans and Mexican immigrants differ in their psychological responses to patterns of lifetime adversity

PONE-D-20-27120R2

Dear Dr. Cooper,

We’re pleased to inform you that your manuscript has been judged scientifically suitable for publication and will be formally accepted for publication once it meets all outstanding technical requirements.

Kind regards,

Stephan Doering, M.D.

Academic Editor

PLOS ONE

---

## [Editor Report · Acceptance letter]

8 Oct 2021

PONE-D-20-27120R2 

Puerto Ricans and Mexican immigrants differ in their psychological responses to patterns of lifetime adversity 

Dear Dr. Cooper:

I'm pleased to inform you that your manuscript has been deemed suitable for publication in PLOS ONE. Congratulations! Your manuscript is now with our production department. 

Kind regards, 

on behalf of

Professor Stephan Doering 

Academic Editor

PLOS ONE